# Quantifying Genetic and Environmental Factors Accounting for Multistage Progression of Precancerous Lesions and Oral Cancer: Applications to Risk-Guided Prevention

**DOI:** 10.3390/cancers17132114

**Published:** 2025-06-24

**Authors:** Donlagon Jumparway, Chiu-Wen Su, Amy Ming-Fang Yen, Yen-Tze Liu, Mu-Kuan Chen, Ko-Jiunn Liu, Pongdech Sarakarn, Sam Li-Sheng Chen

**Affiliations:** 1School of Dentistry, Taipei Medical University, Taipei 110, Taiwan; d204110002@tmu.edu.tw; 2Department of Internal Medicine, National Taiwan University Hospital, Taipei 100, Taiwan; hfn202@gmail.com; 3School of Oral Hygiene, College of Oral Medicine, Taipei Medical University, Taipei 110, Taiwan; amyyen@tmu.edu.tw; 4Big Data Center, Changhua Christian Hospital, Changhua 500, Taiwan; 144084@cch.org.tw; 5Department of Post-Baccalaureate Medicine, College of Medicine, National Chung Hsing University, Taichung 402, Taiwan; 6Department of Family Medicine, Changhua Christian Hospital, Changhua 500, Taiwan; 7Department of Otolaryngology-Head and Neck Surgery, Changhua Christian Hospital, Changhua 500, Taiwan; 53780@cch.org.tw; 8National Institute of Cancer Research, National Health Research Institutes, Tainan 704, Taiwan; kojiunn@nhri.edu.tw; 9Institute of Clinical Pharmacy and Pharmaceutical Sciences, National Cheng Kung University, Tainan 701, Taiwan; 10School of Medical Laboratory Science and Biotechnology, Taipei Medical University, Taipei 110, Taiwan; 11Faculty of Public Health, Khon Kaen University, Khon Kaen 40002, Thailand

**Keywords:** oral cancer, genetic susceptibility, multistate risk assessment model, Markov simulation, screening

## Abstract

Oral squamous cell carcinoma (OSCC) develops through a multistage process driven by both environmental exposures and genetic predisposition. However, current screening programs often consider only environmental exposures, missing the added value of genetic susceptibility. This study proposes an innovative approach by integrating genetic and environmental data into a multistate disease progression model to estimate individual oral cancer risk. Using computer simulations, the study demonstrates that tailored screening strategies can significantly reduce oral cancer incidence, particularly among high-risk groups. These findings offer a new framework for precision prevention and could help guide the development of more effective screening strategies in public health.

## 1. Introduction

Oral cancer, particularly oral squamous cell carcinoma (OSCC), is a multistage and multifactorial disease, strongly influenced by genetic susceptibility and a combination of environmental risk factors [1,2,3]. Exposure to risk factors such as smoking, alcohol consumption, betel quid chewing, and human papillomavirus (HPV) infection, combined with underlying genetic alterations, increases the risk of developing OSCC. The disease typically progresses through a series of stages, from normal mucosa to oral potentially malignant disorders (OPMDs) (such as leukoplakia, erythroplakia, submucous fibrosis, and oral lichen planus), eventually leading to invasive carcinoma [4,5]. This progression involves both environmental factors together with extensive molecular alterations at the genomic, transcriptomic, proteomic, and metabolomic levels. For instance, studies have demonstrated that specific genetic polymorphisms interact with environmental factors to increase OSCC risk. A study from Taiwan has shown that CD44 single nucleotide polymorphisms (SNPs) interact with betel quid chewing and smoking, significantly elevating oral cancer susceptibility [6]. Similarly, mutations in MACC1 have been associated with lymph node metastasis in OSCC patients who chew betel quid [7]. A recent study has unveiled the possibility of identifying high-risk of oral cancer subjects based on genetic risk scores and betel quid [8]. These findings highlight the importance of integrating genetic and environmental data to fully understand the underlying mechanisms driving OSCC.

Due to the complexity and heterogeneity of OSCC, traditional single-factor studies fail to capture the complexity of disease progression. A systems biology approach plays a crucial role in integrating diverse data to form a comprehensive understanding of OSCC development underlying disease progression. Thus, a synthesis approach that accounts for the multistage and multifactorial nature of OSCC is needed for integrating gene-environment interactions and tumor progression mechanisms, enabling the construction of predictive models to assess individual risk based on both genetic predisposition and environmental exposure.

Previous studies elucidated the significant effect of betel quid and cigarette smoking on the multistate progression from disease-free to leukoplakia and leukoplakia to erythroleukoplakia [9,10]. The genetic and epigenetic were considered to play a role in developing oral cancer as well [11,12]. Some studies have identified distinct molecular signatures that differentiate normal, pre-malignant, and malignant tissues [13,14,15]. One of the most common types of genetic variations in the human genome that are associated with genetic susceptibility to cancer is single nucleotide polymorphisms (SNPs), especially when they happen in genes that regulate DNA mismatch repair, cell cycle regulation, metabolism, and immunity [16]. It has been documented as a predictive factor for a high OSCC risk [17].

Although these genetic and molecular alterations play important roles in the development of oral cancer, most oral cancer screenings thus far have been implemented among at-risk populations, determined by age and/or history of environmental risk exposure [18,19,20]. It seems impractical to screen and examine the genetics among population levels. It is of interest to propose an innovative alternative integrated risk classification for oral cancer screening.

In this study, we attempt to construct a synthesis-driven, multi-stage, and multi-factorial integrative model by integrating genetic and epidemiological information based on current evidence to facilitate the risk stratification model. This allows for individual risk stratification and personalized prevention strategies that go beyond traditional risk assessments based on environmental exposure or genes alone for the population. The proposed model was applied to oral cancer screening together with health promotion as an illustration.

## 2. Materials and Methods

### 2.1. Data Source

As oral cancer is a multifactorial and multistate progressive disease, it stands to reason that various factors may influence the transitions between different stages of oral carcinogenesis, including smoking, betel nut chewing, alcohol consumption, human papillomavirus (HPV) infection, genetic polymorphisms, and epigenetic modifications, as shown in Figure 1. The comprehensive systematic review was initially conducted to provide an in-depth integration of the existing literature on risk factors associated with different stages of disease progression.

A comprehensive literature search was conducted from 2009 to 2024 across multiple databases, including PubMed, ScienceDirect, and Google Scholar. The search utilized MeSH terms such as “polymorphisms and oral cancer”, “polymorphisms and OSCC”, “miRNA and oral cancer”, “miRNA and OSCC”, “miRNA and oral potentially malignant disorder”, and “miRNA and leukoplakia”. In addition to electronic searches, a manual review was performed to identify studies focusing on genetic polymorphisms linked to oral cancer progression and the expression of miRNAs in specific oral potentially malignant disorders (OPMDs), such as leukoplakia and precancerous lesions.

By systematically evaluating and integrating findings from multiple studies, the key determinants driving disease development and facilitating transitions across various disease states can be identified. The relative risks associated with environmental, genetic, and epigenetic factors were extracted from the literature as presented in Table 1.

### 2.2. Multistate and Multifactorial Natural History Model of Precancerous and Cancer of Oral Cavity

We constructed a multistate and multifactorial disease natural history that can be delineated as follows: normal → leukoplakia → erythroleukoplakia → oral cancer, superimposed with state-specific factors in each state transition (Figure 1). The state-by-state transition rates per year were obtained from Yen’s study. The baseline hazards of each annual transition rate for subjects with occasional use of betel nut (the reference group) were 0.0014, 0.01917, and 0.1428 per year from normal mucosa to leukoplakia, leukoplakia to erythroleukoplakia, and erythroleukoplakia to oral cancer, respectively [10]. Betel quid and cigarette smoking would contribute to the transformation of the disease from normal to leukoplakia and leukoplakia to erythroleukoplakia.

For developing a personalized risk assessment model, we utilize three regression models to derive transition risk scores. The composite score is then calculated by combining these three transition risk scores, with different weights assigned to each. The weights assigned to each transition were determined based on the relative values derived from the logarithm of the baseline rates for the three transitions.

### 2.3. Computer Simulation

A Markov simulation model was further applied to the cohort with distribution of risk profiles the same as a high-risk group for oral screening in Taiwan. We created the hypothetical cohorts with the integration of environmental and genetic risk factors; afterward, based on the natural history of oral cancer by the four-state Markov model, we began with normal to leukoplakia, leukoplakia to erythroleukoplakia, and erythroleukoplakia to oral cancer. Accordingly, the cohort was stratified into distinct risk groups based on the decile distribution of the composite risk score described above.

We evaluated the outcomes of a screening program implemented with varying inter-screening intervals, either alone or in combination with a health education intervention aimed at promoting cessation of betel quid chewing and cigarette smoking among screening participants. The projected effectiveness was compared to a non-intervention control scenario. These were conceptually shown in Figure 2.

We simulated a hypothetical cohort of 1,000,000 individuals subjected to various inter-screening intervals—annual, biennial, triennial, quinquennial, and decennial—either alone or in combination with a health education intervention targeting cessation of betel quid chewing and cigarette smoking. The model assumed an attendance rate of 60%, a treatment compliance rate of 80% upon detection, and a treatment efficacy of 50% for pre-malignant lesions. All analyses were conducted using SAS software, version 9.4 (SAS Institute Inc., Cary, NC, USA).

## 3. Results

### 3.1. Literature-Review-Derived Odds Ratios Associated with Environmental Factors and Genetic Susceptibility

Table 1 presents the summarized effects of environmental factors on both transitions from normal mucosa to leukoplakia and leukoplakia to erythroleukoplakia. Previous findings show that betel nut chewing is strongly linked to an increased risk of leukoplakia. The risk raised with daily use, reaching a rate ratio of 10.21 for those consuming over 20 pieces per day. Smoking also increased the risk, to a lesser extent, in a dose-dependent manner. The highest rate ratio for smoking was 3.59 among those using 21–30 sticks per day. Moderate to heavy betel nut use also raised the risk of progression from leukoplakia to erythroleukoplakia. This suggests a role in malignant transformation. In contrast, smoking was not found to significantly affect this progression.

The effects of HPV and micro-RNA on the transitions from normal mucosa to leukoplakia are summarized in Table 2. Table 2 shows a strong association between HPV infection and OPMDs, with an odds ratio of 3.87 (95% CI: 2.87–5.21) compared to healthy controls. Four downstream-regulated miRNAs were significantly associated with OPMDs. Specifically, miR-3614-5p (OR: 1.78; 95% CI: 1.21–2.61) and miR-10b-5p (OR: 1.83; 95% CI: 1.19–2.81) showed lower expression levels in OPMD patients compared to healthy controls. Additionally, miR-215-5p (OR: 2.73; 95% CI: 1.19–2.24) and miR-182-5p (OR: 1.51; 95% CI: 1.02–2.24) were also significantly linked to the presence of OPMDs.

The genetic susceptibility (Table 3) and repaired genes (Table 4) were considered to be involved in the progression of oral cancer. A broad range of genetic and molecular alterations were found to be significantly associated with oral cancer progression. Polymorphisms in genes such as PAI-1, ACE, TIMP-2, BRCA1, COL9A1, NOTCH1, HSPA13, NFκB1, and P53 showed notably increased odds ratios (ORs), suggesting their potential role in susceptibility. For instance, individuals with the 4G/4G genotype of PAI-1 had a fivefold increased risk (OR: 5.00; 95% CI: 1.32–8.92), while the G/G genotype of TIMP-2 showed an OR of 26.33 (95% CI: 12.39–55.95), highlighting strong associations. Epigenetic changes were also observed, with methylation of miR-137 associated with a fourfold increased risk (OR: 4.80; 95% CI: 1.23–18.82).

Several single nucleotide polymorphisms (SNPs) in long non-coding RNAs (H19, MALAT1), tumor suppressors (P53), and repair genes (ERCC5, hOGG1) further support a polygenic risk profile (Table 3).

Among the downregulated miRNAs, miR-3614-5p (OR: 6.09; 95% CI: 1.76–21.04) and miR-10b-5p (OR: 3.91; 95% CI: 2.02–7.57) showed strong associations with oral cancer. miR-431-5p was significantly less expressed in oral cancer and associated with a lower risk (OR: 0.40; 95% CI: 0.21–0.76), suggesting a potential protective role. Other miRNAs, including miR-182-5p, miR-215-5p, miR-7-5p, miR-486-3p, and miR-4707-3p, showed non-significant associations (Table 4).

### 3.2. Multistate Risk Score Developments

We applied a proportional hazard form to account for the cumulative effect of covariates expressed by the exponential of each value of the covariate weighted by the corresponding regression coefficient multiplied by the baseline hazard function following Andersen’s method [34]. The baseline hazard function reflects the effect of each covariate in the reference group. Such a modeling framework has been applied in the previous study [9]. In the light of the recognized risk factors and their corresponding effect sizes shown in Table 2, we calculated a series of transition risks from normal to leukoplakia (λ_12_), from leukoplakia to erythroleukoplakia (λ_23_), and from erythroleukoplakia to oral cancer (λ_34_), associated with the corresponding relevant risk factors in the proportional hazard form expressed as follows:
$$\begin{array}{cc} \lambda_{12}=\lambda_{120}\times exp & \left(0.9361\times (Betel\;nut:1\;–10\;pieces\;per\;day\right)+1.7901\times \left(Betel\;nut:11–20\;pieces\;per\;day\right)\\ & +\;2.3234\times \left(Betelnut:20+pieces\;per\;day\right)+0.7324\times \left(Smoking:1–10\;sticks\;per\;day\right)\\ & +1.0428\times \left(Smoking:11–20\;sticks\;per\;day\right)+1.2782\times \left(Smoking:20+sticks\;per\;day\right)\\ & +1.35\times \left(HPV\right)+0.5766\times (miR\_3614)+0.6043\times (miR\_10b)+1.0043\times (miR\_215)\\ & +0.4121\times (miR\_182)) \end{array}$$
$$\begin{array}{cc} \lambda_{23}=\lambda_{230}\times exp & \left(0.5128\times \text{(}Betel\;nut:1–10\;pieces\;per\;day\right)\\ & +\;1.2092\times \left(Betel\;nut:11–20\;pieces\;per\;day\right)\\ & +\;1.2585\times \left(Betel\;nut:20+pieces\;per\;day\right)+0\times \left(Smoking:1–10\;sticks\;per\;day\right)\\ & +\;0.2469\times \left(Smoking:11–20\;sticks\;per\;day\right)\\ & +\;0.2469\times \left(Smoking:20+sticks\;per\;day\text{)}\right) \end{array}$$
$$\begin{array}{cc} \lambda_{34}=\lambda_{340}\times exp & \left(1.61\times \left(PAI-1\right)+2.22\times \left(ACE\;intron16\right)+3.27\times \left(TIMP-2\right)+0.3506\times \left(hOGG1\right)\right)\\ & +0.3507\times \left(ERCC5\right)+1.5686\times \left(miR-137\right)+0.4187\times \left(TIMP3\right)\\ & +0.4947\times (COL9A1)+0.4750\times (BRCA1)+1.1663\times (NOTCH1)\\ & +0.8109\times (HSPA13)+0.4574\times (NF\kappa B1/rs28362491)\\ & +0.4824\times (NF\kappa B1/rs72696119)+0.5306\times (H19/rs217727)\\ & +0.4252\times (H19/rs2839701)+0.3853\times (FAT1)+-0.7885\times (MALAT1)\\ & +0.4947\times (RYR2)+0.6312\times (P53)+1.3635\times (miR\_10b)+1.8066\times (miR\_3614)\\ & +-0.9663\times (miR\_431)+-0.3711\times (miR\_486)+0.5306\times (miR\_182)\\ & +0.7885\times (miR\_215)+0.4947\times (miR\_7)+-0.2357\times (miR\_4707)) \end{array}$$

All these estimated regression baselines are converted to transition rates.

### 3.3. Decile Risk Spectrums of Multistate Transitions

Based on the simulated cohort, individuals in higher-risk deciles show a greater burden of betel nut chewing, smoking, and HPV infection (Table 5). Heavy chewing (≥11 pieces/day) increases substantially. Consumption of 11–20 pieces per day increases from 23.2% in the 1st decile to 39.1% in the 10th decile, while consumption of 20 or more pieces per day rises from 7.2% to 17.4%. The proportion of individuals smoking 11–20 sticks per day increases from 0.1% in the 1st decile to 18.7% in the 10th decile, while those smoking 20 or more sticks per day increase from 0.0% to 10.0%. HPV positivity increases consistently across risk deciles, from 4.9% in the 1st decile to 32.5% in the 10th decile.

In Table 6, the progression from erythroleukoplakia to oral cancer, along with the corresponding lifetime risk of oral cancer, is stratified by risk decile, and the average risk score for each decile is also reported. The 1st group, representing the lowest-risk individuals, had the lowest lifetime risk (362 per 100,000), while the 10th group, representing the highest-risk individuals, had a lifetime risk of 24,523 per 100,000. This pattern is consistent with the mean sojourn time from erythroleukoplakia to oral cancer, which decreases as lifetime risk increases. Mean sojourn time (MST) is the average duration of the stay in erythroleukoplakia. The sojourn time is often assumed to follow an exponential distribution. In the 1st risk group, progression takes over 45.5 years, whereas in the 10th group, it occurs within approximately 1 to 2 months. The shorter the MST, the more likely the progression to oral cancer. The relative risk (RR) indicates that the 10th group has a 7.33-fold increased risk, while the 1st group has an 89% lower risk of progression compared to the average-risk group (the 5th decile).

In Figure 3A, we observed a 46% reduction in oral cancer incidence among screened subjects compared to the non-screened group for annual screening, 45% for triennial, 44% for quinquennial, and 40% for decennial screening programs in the average-risk group. The reduction decreased with longer screening intervals. These results were estimated based on assumptions of 50% efficacy of treatment for pre-malignancy, a 60% attendance rate, and an 80% compliance rate with treatment once detected. These trends also indicate that more frequent screening is more effective in reducing incidence, particularly in high-risk groups. Longer intervals are associated with less reduced effectiveness.

The incidence rate reduction showed a consistent trend across different risk groups, regardless of the inter-screening interval. When an educational program promoting the cessation of betel quid chewing and cigarette smoking was incorporated into the screening program, an additional reduction in incidence was observed. In the 50th risk group, incidence reductions were 48% for both 1- and 3-yearly screening, 47% for 5-yearly, and 44% for 10-yearly screening among those who underwent screening compared to the control group, which reflects an additional 2% to 6% reduction attributable to the health education intervention. The added benefit of health education was more pronounced in higher-risk groups, as illustrated in Figure 3B.

## 4. Discussion

This study has demonstrated the feasibility of integrating both environmental and genetic risk factors for risk stratification in the implementation of population-based oral cancer screening. In addition to evaluating the effectiveness of oral cancer screening for reducing oral cancer incidence, which varies by screening interval and risk groups, we also considered the implementation of the health education program targeting betel quid chewing and cigarette smoking cessation as part of the screening initiative. All scenarios have been estimated under determined parameters, including a 60% attendance rate, 80% compliance to treatment once detected, and 50% treatment efficacy for pre-malignancy.

We classified the cohort into deciles based on risk, where higher deciles corresponded to greater lifetime risk of oral cancer. The mean sojourn time (MST) showed an inverse pattern. Higher-risk groups had shorter MSTs, while lower-risk groups had longer MSTs. As expected, shorter screening intervals were associated with greater effectiveness in reducing oral cancer incidence. At the average risk level (50th percentile), the incidence reduction was highest with annual screening (46%), followed by 3-yearly (45%), 5-yearly (44%), and 10-yearly (40%) screening. When a health education program was introduced, the effectiveness increased slightly, with improvements of 2% for annual, 3% for both 3-yearly and 5-yearly, and 6% for 10-yearly screening intervals.

We compared the incidence reduction estimated in our study with that of a study conducted in India. A decline in incidence was demonstrated in the cluster-randomized trial in India, which implemented four screening rounds with triennial intervals. They reported a 38% reduction in oral cancer incidence [35]. In Taiwan, an organized, community-based nationwide oral cancer screening program has been implemented since 2004, targeting individuals aged over 18 who smoke cigarettes, chew betel quid, or both. Among more than 2 million participants, biennial oral visual inspection-based screening for high-risk groups has been shown to reduce oral cancer incidence by 17% after adjustment for self-selection bias [19].

Our results show that the incidence reduction from triennial screening in the 50th percentile risk group is 7% and 10% higher than that reported in a cluster-randomized trial conducted in India, without and with the inclusion of a health education program, respectively. This greater reduction may suggest that our integration of both environmental and epigenetic or genetic risk factors contributes to a more accurate classification of oral cancer risk. In contrast, the population-based oral cancer screening program in Taiwan, which targets individuals with high-risk oral habits (betel quid chewing and cigarette smoking) using a biennial screening interval, demonstrated a 17% greater reduction in oral cancer incidence among the screened group compared to the non-screened group. The effectiveness of oral cancer screening has been demonstrated using a three-state natural history model of oral cancer. In that study, approximately a 30% reduction in incidence was observed among individuals exposed to all three risk factors (betel quid, smoking, and alcohol) with 10-yearly screening [9]. Using a four-state model, our results show that, at the 50th risk group, the reduction in oral cancer incidence is 10% higher compared to the estimate for individuals exposed to all three risk factors under the same 10-year screening interval. It may not be straightforward to compare and interpret the efficacy of incidence reduction directly. It should be noted that the lower reduction in oral cancer incidence reported previously may be due to the lower attendance rate and the shorter follow-up period, for example, 4.5 years in the Taiwanese oral cancer screening program [19]. As a result, the observed incidence reduction in the Taiwanese population-based oral cancer screening program is lower than our estimated results.

From an epidemiological perspective, primary prevention is one of the key strategies for reducing the incidence of oral cancer. This approach focuses on identifying risk factors and either minimizing exposure to them or strengthening host defenses against disease, thereby lowering disease occurrence at the population level. HPV infection as one of the environmental risk factors can be prevented through vaccination [36]. Cessation of exposure to risk factors, including tobacco smoking and use of betel nut with or without tobacco, will contribute to significant reductions in the risk of oral cancer, which has been established in the IARC working group [37]. The effect of smoking and betel nut chewing cessation by health education was considered in our analysis. The secondary prevention, the oral cancer screening, could be considered as a promising approach to detect, prevent, and terminate a transformation from precancerous to oral malignancy. Incorporating genetic information into risk stratification may enhance oral cancer screening beyond the use of environmental risk factors alone. Risk profiling can support the development of both tailored primary and secondary prevention strategies.

Based on a progressive continuum of leukoplakia lesions classified by Bouquot and Whitaker [3], our natural history model from leukoplakia to oral cancer was adopted from the four-state progressive model from Yen et al. [10]. As the risk of having malignant transformation to oral cancer also increases in parallel with the severity of the subtypes of OPMD from leukoplakia to erythroleukoplakia, such a linear progression to oral cancer may not be unreasonable. Such a multistate Markov process follows such a linear progression, but cumulative exposure to state-specific risk factors responsible for each transition may account for a nonlinear oral carcinogenesis. The current piecewise multistate risk analysis applied to each state transition may not violate the assumption of this linear progression.

There are several limitations in this study. First, our data were retrieved from literature reviews, particularly concerning candidate susceptibility genes for oral cancer, some of which were not derived from studies conducted in Taiwanese populations. Second, we included only betel quid chewing and cigarette smoking as risk factors, based on strong evidence supporting their roles in the multistate progression of oral cancer in Taiwan. However, other potential risk factors, such as alcohol consumption, may also play a role [38,39] in the risk development of oral cancer. Third, our multistate model from leukoplakia to oral cancer was adopted from the four-state progressive model [10]. Based on this natural history model, the joint effects of major environmental factors such as betel nut chewing and smoking were already considered in the multistage progression model. It should be noted that state-specific covariates responsible for each step of transition have been assigned in light of the findings acknowledged in the previous studies. For example, the major effect of smoking lies on the occurrence of leukoplaka, although smoking may still play a minor role in the malignant transformation. HPV is responsible for the occurrence of leukoplakia. The effects of miRNA on different states were assigned according to the outcome comparisons selected in the previous studies. In our current study, we used SNP information from individual studies separately, as this better reflects the individual genetic risk. However, these environmental factors may also interact with SNPs. Four, using effect size estimates from individual studies represents a limitation of our model, and future research should consider synthesizing pooled estimates from systematic reviews or meta-analyses, which might enhance the robustness and generalizability of our model. Lastly, this simulation was based on the context of oral cancer screening in Taiwan and may not be generalizable to other countries, such as the United States, where no evidence has been found to support oral cancer screening even among high-risk populations [40].

## 5. Conclusions

In conclusion, oral cancer risk is influenced by a combination of genetic and environmental factors, such as betel quid chewing and cigarette smoking. Our study demonstrated the feasibility of incorporating these risk factors to develop a risk assessment model for accommodating a multistate oral cancer progression framework applicable across all age groups. Simulations based on different prevention programs showed varying effectiveness across risk groups. Introducing a health education program for cessation of betel quid chewing and cigarette smoking within an oral cancer screening program could provide an additional 2 to 6% reduction in incidence compared to a universal program. This state-of-the-art Markov simulation model is applicable for evaluating risk-oriented oral cancer screening strategies and can support evidence-based decision-making in screening policy development.

## Figures and Tables

**Figure 1 cancers-17-02114-f001:**
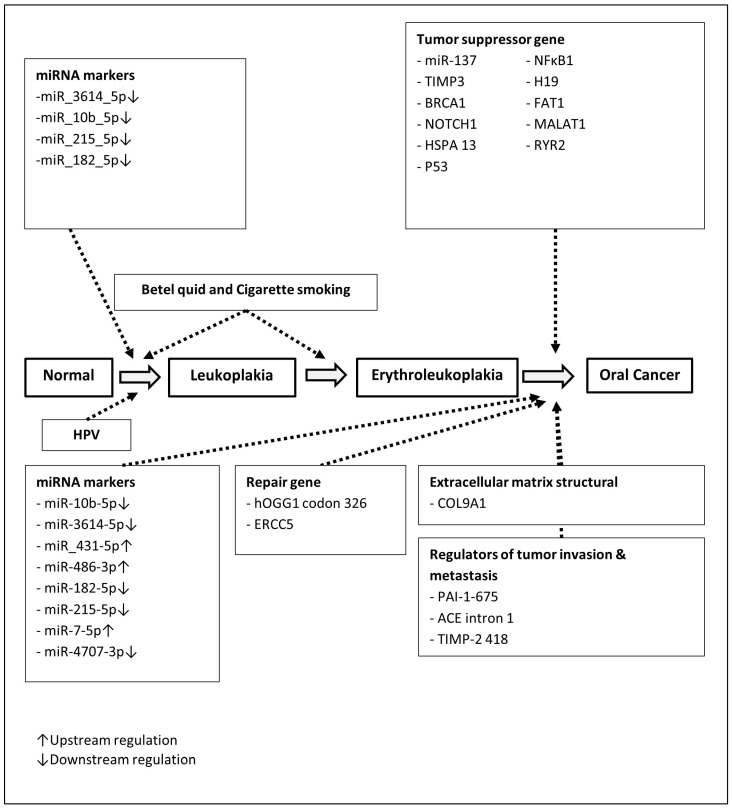
Betel quid, cigarette smoking, and genetic and molecular alterations in oral cancer progression.

**Figure 2 cancers-17-02114-f002:**
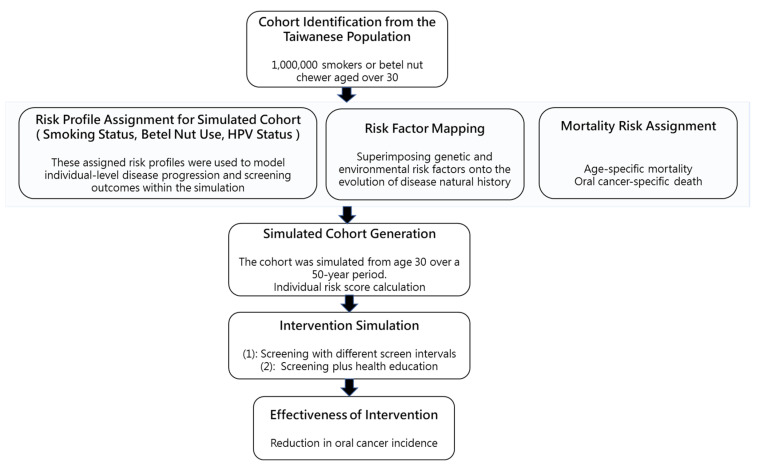
Assessing the impact of screening and health education on oral cancer progression by Markov simulation model.

**Figure 3 cancers-17-02114-f003:**
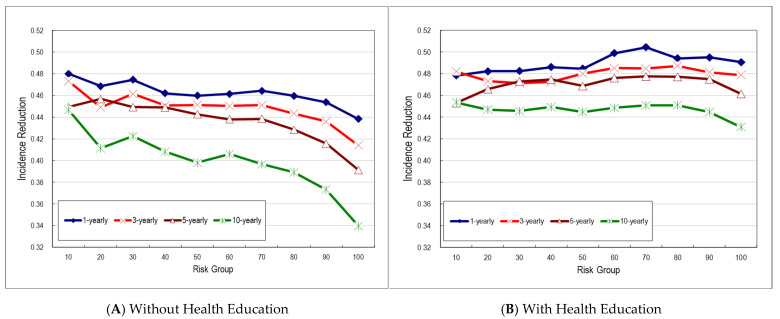
Effectiveness for reducing oral cancer incidence for screening strategies with different screen-intervals compared to no screening.

**Table 1 cancers-17-02114-t001:** The effects of environmental factors on the multistep progression of oral potentially malignant disorder.

Variables	Coefficients	Rate Ratio	Reference
The effect of betel nut chewing on incidence of leukoplakia			
Occasional use	0	1.0	[10]
1–10 pieces per day	0.936	2.55 (1.91–3.41)	
11–20 pieces per day	1.790	5.99 (4.25–8.45)	
20+ pieces per day	2.323	10.21 (7.53–13.84)	
The effect of smoking on incidence of leukoplakia			
Occasional use	0	1.0	
1–10 sticks per day	0.732	2.08 (1.26–3.42)	
11–20 sticks per day	1.040	2.83 (1.74–4.59)	
21–30 sticks per day	1.278	3.59 (2.16–5.95)	
The effect of betel nut chewing on transition from leukoplakia to erythroleukoplakia			
Occasional use	0	1.0	
1–10 pieces per day	0.513	1.67 (0.84–3.33)	
11–20 pieces per day	1.209	3.35 (1.65–6.83)	
20+ pieces per day	1.258	3.52 (1.80–6.86)	
The effect of smoking on transition from leukoplakia to erythroleukoplakia			
Occasional use	0	1.0	
1–10 sticks per day	0	1.00 (0.42–2.39)	
11–20 sticks per day	0.199	1.22 (0.53–2.83)	
21–30 sticks per day	0.199	1.22 (0.51–2.93)	

**Table 2 cancers-17-02114-t002:** Odds ratios of oral potentially malignant disorder associated with HPV and four-miRNA panel.

Genes	OPMD N (%)/Mean (SE)	Healthy N (%)/Mean (SE)	OR	95% CI	References
HPV					[21]
Negative	601 (63%)	586 (87%)			
Positive	355 (37%)	89 (13%)	3.87	2.87–5.21	
miR-3614-5p	−8.5 (3.9)	−9.2 (2.8)	1.78	1.21–2.61	[22]
miR-10b-5p	−8.5 (7.1)	−10.0 (2.8)	1.83	1.19–2.81	
miR-215-5p	−2.85 (1.6)	−3.0 (1.7)	2.73	1.19–2.24	
miR-182-5p	−4.5 (3.5)	−5.3 (2.8)	1.51	1.02–2.24	

**Table 3 cancers-17-02114-t003:** Evidence of genetic and molecular alterations associated with progression of oral cancer.

Genes	Cases N (%)	Control N (%)	OR	95% CI	References
PAI-1–675					[23]
5G/4G or 5G/5G	59 (56.7)	75 (70.8)	1.00		
4G/4G	45 (43.3)	31 (29.2)	5.00	1.32–8.92	
ACE intron 16					
D/I or D/D	130 (81.3)	144 (94.1)	1.00		
I/I	30 (18.7)	9 (5.9)	9.16	1.14–73.50	
TIMP-2–418					
C/C and G/C	66 (41.8)	159 (94.6)	1.00		
G/G	92 (58.2)	9 (5.4)	26.33	12.39–55.95	
hOGG1 codon 326 (rs1052133)					[24]
CG/GG	482 (77.7)	516 (83.2)	1.00		
CC	138 (22.3)	104 (16.8)	1.42	1.08–1.89	
ERCC5 (rs751402)					[25]
CC	98 (41.0)	167 (49.7)	1.00		
CT/TT	141 (59.0)	169 (50.3)	1.42	1.02–1.99	
miR-137					[26]
Unmethylated	78 (78.8)	96 (97.0)	1.00		
Methylated	21 (21.2)	3 (3.0)	4.80	1.23–18.82	
TIMP3 (rs9862)					[27]
CC	192 (25.7)	414 (34.5)	1.00		
CT/TT	555 (74.3)	786 (65.5)	1.52	1.24–1.86	
BRCA1 (rs2070833)					[8]
CC	237 (53.0)	374 (64.5)	1.00		
CT/TT	210 (47.0)	206 (35.5)	1.61	1.25–2.07	
COL9A1 (rs550675)					
CC	187 (41.8)	314 (54.1)	1.00		
CT/TT	260 (58.2)	266 (45.9)	1.64	1.28–2.11	
NOTCH1 (rs139994842)					
GG	401 (89.7)	560 (96.6)	1.00		
AG/AA	46 (10.3)	20 (3.4)	3.21	1.87–5.51	
HSPA13 (rs2822641)					
GG	386 (86.4)	542 (93.4)	1.00		
GT/TT	61 (13.6)	38 (6.6)	2.25	1.47–3.45	
NFκB1 (rs28362491)					[28]
Ins/Ins + Ins/Del	321 (76.2)	393 (81.4)	1.00		
Del/Del	100 (23.8)	90 (18.6)	1.58	1.10–2.26	
NFκB1 (rs72696119)					
CC + CG	318 (75.53)	388 (81.0)	1.00		
GG	103 (24.5)	91 (19.0)	1.62	1.14–2.32	
H19 (rs217727)					[29]
CC + TC	380 (85.6)	911 (92.6)	1.00		
TT	51 (11.5)	73 (7.4)	1.70	1.16–2.49	
H19 (rs2839701)					
CC + CG	393 (88.5)	909 (92.4)	1.00		
GG	51 (11.5)	75 (7.6)	1.53	1.05–2.24	
FAT1 (rs28647489)					[30]
GA + AA	285 (79.2)	412 (84.8)	1.00		
GG	75 (20.8)	74 (15.2)	1.47	1.03–2.09	
MALAT1 (rs3200401)					[31]
CC	948 (70.2)	807 (67.3)	1.00		
CT/TT	402 (29.8)	392 (32.7)	0.78	0.63–0.96	
RYR2 (rs12594)					[32]
AA/AG	509 (90.5)	315 (94.9)	1.00		
GG	53 (9.5)	17 (5.1)	1.93	1.10–3.39	
P53					[33]
Negative	330 (54.2)	421 (69.0)	1.00		
Positive	279 (45.8)	189 (31.0)	1.88	1.39–2.56	

**Table 4 cancers-17-02114-t004:** Odds ratios of oral cancer associated with eight-miRNA panel.

Genes	Oral Cancer Mean (SE)	Healthy Mean (SE)	OR	95% CI	References
miR-10b-5p	−8.2 (3.9)	−10.0 (2.8)	3.91	2.02–7.57	[22]
miR-3614-5p	−9.0 (3.5)	−9.2 (2.8)	6.09	1.76–21.04	
miR_431-5p	−5.8 (4.9)	−3.1 (2.8)	0.40	0.21–0.76	
miR-486-3p	−8.9 (4.9)	−6.0 (4.5)	0.69	0.47–1.02	
miR-182-5p	−5.0 (4.9)	−5.3 (2.8)	1.70	0.69–4.20	
miR-215-5p	−3.0 (2.6)	−3.0 (2.7)	2.20	0.56–8.66	
miR-7-5p	−3.8 (2.0)	−4.9 (3.9)	1.64	0.69–3.90	
miR-4707-3p	−7.0 (4.4)	−5.2 (4.4)	0.79	0.39–1.57	

**Table 5 cancers-17-02114-t005:** Distribution of betel nut chewing, smoking, and HPV infection among simulated cohort across oral cancer risk deciles.

Characteristics	Risk Decile									
	1	2	3	4	5	6	7	8	9	10
Betel Nut Chewing										
Occasional use	36.9%	28.9%	26.4%	24.8%	23.6%	22.3%	21.2%	20.1%	19.0%	17.0%
1–10 pieces per day	32.7%	32.5%	31.7%	31.2%	30.7%	30.2%	29.5%	28.9%	28.0%	26.5%
11–20 pieces per day	23.2%	28.8%	30.9%	32.1%	33.1%	34.1%	35.2%	36.3%	37.3%	39.1%
20+ pieces per day	7.2%	9.9%	11.0%	11.9%	12.6%	13.3%	14.1%	14.6%	15.8%	17.4%
Smoking										
No Smoking	95.0%	90.2%	86.9%	83.6%	80.2%	76.6%	72.7%	68.3%	62.1%	53.7%
1–10 sticks per day	4.9%	9.0%	11.1%	12.9%	14.6%	16.2%	17.3%	18.0%	19.1%	17.6%
11–20 sticks per day	0.1%	0.7%	1.5%	2.5%	3.7%	5.1%	7.1%	9.6%	12.7%	18.7%
20+ sticks per day	0.0%	0.2%	0.5%	0.9%	1.4%	2.1%	3.0%	4.2%	6.2%	10.0%
HPV Infection										
No	95.1%	91.2%	88.9%	86.7%	84.8%	82.8%	80.7%	78.0%	74.8%	67.5%
Yes	4.9%	8.8%	11.1%	13.3%	15.2%	17.2%	19.3%	22.0%	25.2%	32.5%

**Table 6 cancers-17-02114-t006:** Mean risk score, lifetime risk and mean sojourn time from erythroleukoplakia to oral cancer by risk decile.

Characteristics	Risk Decile									
	1	2	3	4	5	6	7	8	9	10
Mean risk score	3.4	4.8	5.6	6.3	6.9	7.5	8.2	8.9	9.9	11.8
Lifetime risk (per 10^5^)	362	925	1576	2360	3348	4753	6698	9551	13239	24523
Relative risk	0.11	0.28	0.47	0.71	1.00	1.42	2.00	2.85	4.25	7.33
MST of from eleukoplakia to oral cancer (year)	45.5	18.7	10.8	6.9	4.5	3.1	2.0	1.3	0.7	0.1

## Data Availability

Data supporting this study are available from the corresponding author on reasonable request.

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
