# Peer review of "Quantifying Genetic and Environmental Factors Accounting for Multistage Progression of Precancerous Lesions and Oral Cancer: Applications to Risk-Guided Prevention"

_cancers, 2025, doi:10.3390/cancers17132114_

Round 1
Reviewer 1 Report
Comments and Suggestions for Authors
The study is methodologically sound, structured in a complex but coherent way, with a clear presentation of the data and an effective interpretation of the results.
The proposal of a risk stratification model that integrates environmental factors and genetic predisposition represents a relevant contribution both in the clinical field and for individual risk management. The results obtained, as underlined by the authors, provide a solid rationale for the implementation of primary and secondary prevention strategies
Only minor comments: The materials and methods section should be moved and placed before the results section.
Line 228:The mean sojourn time (MST) showed an inverse pattern. How could this result be explained? Do the authors have any hypotheses?
Author Response
Comments 1: The study is methodologically sound, structured in a complex but coherent way, with a clear presentation of the data and an effective interpretation of the results.
The proposal of a risk stratification model that integrates environmental factors and genetic predisposition represents a relevant contribution both in the clinical field and for individual risk management. The results obtained, as underlined by the authors, provide a solid rationale for the implementation of primary and secondary prevention strategies.
Response 1: Thank you for reviewer’s kind words.
Comments 2: Only minor comments: The materials and methods section should be moved and placed before the results section.
Response 2: Agreed. We have moved the materials and methods section before the results section.
Comments 3: Line 228:The mean sojourn time (MST) showed an inverse pattern. How could this result be explained? Do the authors have any hypotheses?
Response 3: The mean sojourn time is the average duration of staying in erythroleukoplakia. The sojourn time is often assumed to follow an exponential distribution. The MST for the average-risk individual (the 5th risk decile) is expected to take approximate 4.5 years to progress from erythroleukoplakia to oral cancer. However, for the extremely high-risk group, the progression would be more rapid and the MST is significantly shorter than 0.1 year. The shorter the MST, the more likely the progression to oral cancer. This point has been addressed in the revised manuscript (Page 11, Line 273-277)
Reviewer 2 Report
Comments and Suggestions for Authors
Figure 2 (Page 13) presents a multistep model of oral cancer progression, describing a linear development from normal mucosa to leukoplakia, erythroleukoplakia, and ultimately to oral cancer. While conceptually informative, the model appears to assume a strictly sequential and linear progression of disease. This assumption oversimplifies the complex and often nonlinear nature of oral carcinogenesis. The authors should provide a stronger biological rationale to support the linear framework proposed.
The derivation of transition risks (λ₁₂, λ₂₃, λ₃₄) using proportional hazards modeling, as shown on page 8—is critical to the validity of the entire analysis. However, the explanation of this modeling framework is currently insufficient. It is essential to describe how the baseline hazards (λ120, λ230, and λ340) were estimated or assumed, whether they were derived from empirical data, literature, or modeled as constants.
Furthermore, the study includes a wide array of risk factors, each assigned to a specific transition step. However, the rationale for assigning individual risk factors to distinct stages of progression is not adequately justified. The biological specificity of these assignments remains unclear, especially considering the potential for overlapping or interacting effects across stages. The model appears to assume independence among factors, yet no evidence or discussion is provided to support this assumption.
According to Tables 1 through 4, each OR appears to be drawn from a single study. This approach introduces considerable uncertainty and undermines the reliability of the model estimates. It is strongly recommended that the authors consult systematic reviews or meta-analyses where available, to obtain more robust and generalizable effect size estimates.
Author Response
Comments 1: Figure 2 (Page 13) presents a multistep model of oral cancer progression, describing a linear development from normal mucosa to leukoplakia, erythroleukoplakia, and ultimately to oral cancer. While conceptually informative, the model appears to assume a strictly sequential and linear progression of disease. This assumption oversimplifies the complex and often nonlinear nature of oral carcinogenesis. The authors should provide a stronger biological rationale to support the linear framework proposed.
Response 1: Thankful for the reviewer’s valuable comments. Bouquot and Whitaker have depicted the biological changes in oral premalignant disorders as a progressive continuum of leukoplakia lesions, each associated with an increasing risk of malignant transformation (Bouquot et al., 1994). Base on their classifications of OPMD, our natural history model from leukoplakia to oral cancer was adopted by four-state progressive model from Yen et al. (Yen et al., 2008). As the risk of having malignant transformation to oral cancer also increases in parallel with the severity of the subtypes of OPMD from leukoplakia to erythroleukoplakia such a linear progression to oral cancer may not be unreasonable. Such a multistate Markov process following such a linear progression but cumulative exposure to state-specific risk factors responsible for each transition may account for a nonlinear oral carcinogenesis. The current piecewise multistate risk analysis applied to each state transition may not violate the assumption of this linear progression. This point has been addressed in the Discussion section (Page 14, Line 365-374)
Comments 2: The derivation of transition risks (λ₁₂, λ₂₃, λ₃₄) using proportional hazards modeling, as shown on page 8—is critical to the validity of the entire analysis. However, the explanation of this modeling framework is currently insufficient. It is essential to describe how the baseline hazards (λ120, λ230, and λ340) were estimated or assumed, whether they were derived from empirical data, literature, or modeled as constants.
Response 2: In the current study, we applied a proportional hazard form to account for the cumulative effect of covariates expressed by the exponential of each value of the covariate weighted by the corresponding regression coefficient multiplied by baseline hazard function following the Andersen’s method (Anderson, 1992). The baseline hazard function reflects the effect of each covariate in the reference group. Such a modeling framework has been applied in several previous studies (Shu et al., 2004; Hsu et al., 2020). (Page 10, Line 219-224). The baseline hazards were obtained from Yen’s study (Yen et al., 2008). The sentence was amended as “ The baseline hazards of each annual transition rate for subjects with occasional use of betel nut (the reference group) were 0.0014, 0.01917, and 0.1428 per year from normal mucosa to leukoplakia, leukoplakia to erythroleukoplakia, and erythroleukoplakia to oral cancer, respective-ly[10].” (Page 3, Line 131-134). The references have been cited.
Andersen PK. Repeated assessment of risk factors in survival analysis. Stat Methods Med Res. 1992;1(3):297-315.
Hsu CY, Hsu WF, Yen AM, Chen HH. Sampling-based Markov regression model for multistate disease progression: Applications to population-based cancer screening program. Stat Methods Med Res. 2020 Aug;29(8):2198-2216.
Shiu MN, Chen TH. Impact of betel quid, tobacco and alcohol on three-stage disease natural history of oral leukoplakia and cancer: implication for prevention of oral cancer. Eur J Cancer Prev. 2004 Feb;13(1):39-45.
Comments 3: Furthermore, the study includes a wide array of risk factors, each assigned to a specific transition step. However, the rationale for assigning individual risk factors to distinct stages of progression is not adequately justified. The biological specificity of these assignments remains unclear, especially considering the potential for overlapping or interacting effects across stages. The model appears to assume independence among factors, yet no evidence or discussion is provided to support this assumption.
Response 3: Our multistate model from leukoplakia to oral cancer was adopted by four-state progressive model from Yen et al. (Yen et al., 2008). Based on this natural history model, the joint effects of major environmental factors such as betel nut chewing and smoking were already considered in multistage progression model. It should be noted that state-specific covariates responsible for each step of transition have been assigned in the light of the findings acknowledged in the previous studies. For example, the major effect of smoking lays on the occurrence of leukoplaka although smoking may still play a minor role in the malignant transformation. HPV is responsible for the occurrence of leukoplakia. The effects of miRNA on different states were assigned according to the outcome comparisons selected in the previous studies. In our current study, we used SNP information from individual studies separately as this better reflects the individual genetic risk. However, these environmental factors may also interact with SNPs. This would be one of limitations in our study (Page 14, Line 381-392).
Comments 4:According to Tables 1 through 4, each OR appears to be drawn from a single study. This approach introduces considerable uncertainty and undermines the reliability of the model estimates. It is strongly recommended that the authors consult systematic reviews or meta-analyses where available, to obtain more robust and generalizable effect size estimates.
Response 4: We appreciate the insightful suggestion from reviewer. We fully agree that using effect size estimates from systematic reviews or meta-analyses would enhance the robustness and generalizability of the model. However, conducting such a synthesis is beyond the scope of the current study, which aims to demonstrate a methodological framework using available data. We acknowledge this as a limitation and have noted it in the revised manuscript (Page 14, Line 393-396).
Reviewer 3 Report
Comments and Suggestions for Authors
This is an interesting epidemiological study on quantifying genetic and environmental factors that lead to premalignant and malignant lesions of the oral cavity with clinical importance in the prevention strategies.
- the fact that it is not including an established risk factor for neoplasia formation of the oral cavity like alcohol consumption is a strong limitation
- as the results are presented before the material and methods, its is quite hard for a reader not familiar with epidemiology methodology to follow.
Author Response
Comments 1:This is an interesting epidemiological study on quantifying genetic and environmental factors that lead to premalignant and malignant lesions of the oral cavity with clinical importance in the prevention strategies. the fact that it is not including an established risk factor for neoplasia formation of the oral cavity like alcohol consumption is a strong limitation
Response 1: Thank you for the valuable comment. Since our illustration is based on the oral cancer prevention context among tobacco and betel nut users in Taiwan, we acknowledge that other factors such as alcohol consumption is also very important. The inability to incorporate all of other possible factors into the model represents a limitation of the study (Page 14, Line 377-381).
Comments 2:as the results are presented before the material and methods, its is quite hard for a reader not familiar with epidemiology methodology to follow.
Response 2: Agreed. We have moved the materials and methods section before the results section.
Round 2
Reviewer 2 Report
Comments and Suggestions for Authors
The authors have answered my concerns. No further comments.
Reviewer 3 Report
Comments and Suggestions for Authors
I should than the authors for adressing my comments.